# Uncovering the Work–Family Interface: The Impact of Facilitators and Stressors on the Health of Farm Women

**DOI:** 10.3390/healthcare11202726

**Published:** 2023-10-13

**Authors:** Gloria Mora-Guerrero, Fernanda Herrera-González, Jorge Constanzo-Belmar, Carolina Alveal-Álamos, Sharon Viscardi

**Affiliations:** 1Departamento de Psicología, Facultad de Ciencias de la Salud, Universidad Católica de Temuco, Campus San Francisco, Manuel Montt 056, Temuco 4813302, Chile; 2Núcleo de Estudios Interculturales e Interétnicos, Universidad Católica de Temuco, Campus San Francisco, Manuel Montt 056, Temuco 4813302, Chile; 3Instituto Interdisciplinario del Agua-Rukako, Universidad Católica de Temuco, Rudecindo Ortega 02950, Temuco 4813302, Chile; 4Programa de Doctorado en Planificación Territorial y Sustentabilidad, Universidad Católica de Temuco, Campus Luis Rivas del Canto, Rudecindo Ortega 03694, Temuco 4813302, Chile; 5Laboratorio de Investigación Interdisciplinaria en Microbiología Aplicada, Departamento de Procesos Diagnósticos y Evaluación, Facultad de Ciencias de la Salud, Universidad Católica de Temuco, Campus San Francisco, Manuel Montt 56, Temuco 4813302, Chile; 6Núcleo de Investigación en Producción Alimentaria, Universidad Católica de Temuco, Rudecindo Ortega 02950, Temuco 4813302, Chile; 7Biotechnology of Functional Foods Laboratory, Camino Sanquilco, Parcela 18, La Araucanía, Padre Las Casas 4850827, Chile

**Keywords:** work–family interface, work–family conflict, work–family facilitation, stressors, gender differences

## Abstract

Work–family interface (WFI) theory has identified many stressors that influence work–family dynamics from the standpoint of employees. However, work–family facilitators, as well as the effects of gender differences and the impact of sociocultural environments that differ from a formal employment situation, have received much less attention. Our research aimed to fill these theoretical gaps by analyzing the facilitators and stressors involved in work–family dynamics and determining their consequences for farm women’s physical, psychological, and social health. We used a qualitative method with a grounded theory design to collect data via semi-structured interviews with 46 farm women from the region of Araucanía in Chile. Our results explain how facilitators, stressors, and outcomes take place in a process of work–family balance that, paradoxically, implies exhausting journeys, a gender-based overload, a risk of diffuse body pain and distress, and a lack of time for personal healthcare and productive autonomy. Addressing these issues requires a comprehensive approach involving improved healthcare infrastructure and services focused on changing the pressures that the farming WFI exerts on rural women.

## 1. Introduction

Over the last three decades, a large amount of research has highlighted that there is a significant relationship between people’s well-being and the dynamics between the domains of their work and of their family. An influential theoretical framework that was generated to explore this topic is called roles theory. Proposed by Greenhaus and Beutell [1], this theory suggests that a conflict emerges when a certain quantity of time or specific behaviors required by one role cause sufficient tension that makes it difficult to fulfill the requirements of the other domain. Frone, Russell, and Cooper [2] extended this model by proposing the work–family interface (WFI), not only enabling the positioning of domain-specific evidence and the outcomes of this conflict but also balancing between work and family dynamics. 

Abundant literature has identified many negative work–family (WF) outcomes, such as difficulties in the development of paternal–maternal functions, depression, negative emotions, poor physical health, self-diagnosed mental health issues, heavy drinking, dissatisfaction with life, work stress, less commitment to a work organization, higher levels of absenteeism, and the desire to change jobs [3,4]. Among the factors that influence such outcomes, several studies have indicated that supervisor/co-worker/family support, work/family time commitments, work dissatisfaction, job/parental overload, perceiving more skill discretion or decision authority at work, and being married could be stressors or facilitators of WF relations [4]. However, almost all these studies have addressed employees’ experiences without considering gender differences or the impact of the characteristics of sociocultural environments that differ from employment at a formal organization.

Women’s work on farms differs from that of being an employee because, on a farm, work, familial labor, and leisure are not separate entities; instead, these activities are often trained and become infused into daily life [5,6]. Farming is commonly referred to by women as a lifestyle in which work and family roles are intertwined, and the boundaries between them are often unclear [5,6,7]. On the other hand, farming women are trained, because of gender-based socialization, to be responsible for domestic and care roles in a social position subordinated to men [8].

Studying the relation between farm work, gender-based familial characteristics, and the WFI is relevant since, according to Putnik et al. [9] and Bragger et al. [10], sociocultural context is a mediator between the stressors and outcomes of WF dynamics. The WFI has implications for mental health, stress, depression, and the abandonment of work among rural women [11], as engaging in physically demanding occupations, such as agricultural work, exposes them to various occupational hazards [12]. Thus, in rural areas, better conditions for nurturing a healthy WF balance for women are urgently needed to improve women’s incomes, nutrition, health, and education for themselves and their families [13]. However, to the best of our knowledge, no studies have been performed that evaluate the WFI facilitators and stressors involved in the experiences of farming women. Our objective was to determine which facilitators or stressors are involved in a strained and/or balanced WF dynamic and analyze their physical, psychological, and social consequences on farm women.

### 1.1. Evidence and Outcomes in the Work–Family Interface

In the 1980s, Greenhaus and Beutell [1] proposed the work–family conflict (WFC) model to explain why a conflict emerges when people play, work, and fulfill family roles that require energy, attitudes, or time that interfere with each other. A relevant advance occurred in the 1990s, when Gutek, Searle, and Klepa [14] added gender to this equation, arguing that it influences women and men differently and that, as a consequence, it is necessary to propose WFC as a construct generated by two bidirectional effects: (a) work-to-family conflict, which occurs when work interferes with family life and is expected more among women because of their greater relevance in the domestic sphere, and (b) family-to-work conflict, which occurs when family life interferes with work and is predicted most among men, who have greater expectations to fulfill in the labor sphere [3].

Also in the 1990s, Frone, Russell, and Cooper [2] proposed the WFI model to position the unique, domain-specific characteristics and outcomes of both types of conflict. In this context, scholars have researched the WFC by addressing, firstly, their negative outcomes [3,4] and, secondly, their experiences identified as stressors [15,16]. More recently, Stoiko, Strough, and Turiano [4] added facilitators as factors that positively influence work-to-family and family-to-work interactions. Nevertheless, although studies have determined statistical relationships between stressors, facilitators, and WFC, they did not address how WF dynamics are experienced differently by women and men in specific cultural contexts [9,10].

In fact, extensive research has pointed out the stressors that predict the interference of the WF balance. Working long hours, having a greater workload pressure, longer commuting times, bringing work home, more contact with work at home, work–family multitasking, less decision latitude and less support at work, having children under the age of five, and time spent on care are all factors associated with greater WFC [4]. On the other hand, perceiving no or low task overload is associated with no WF interference, while overtime and work pace are associated with work-to-family conflict but not with family-to-work conflict in the experiences of Finnish working families [16]. Regarding facilitators, Refs. [4,17] agree that women’s autonomy to make decisions and familial support for care and domestic tasks are key factors in striking a WF balance.

Marra [18], Lemos, Barbosa, and Monzato [19], and Putnik et al. [9] have criticized the fact that the vast majority of the existing knowledge in this field predominantly centers around white women from middle-class backgrounds in developed countries. Consequently, the overall findings fail to adequately address the circumstances faced by women with limited decision-making power, particularly those situated in developing countries. These authors have proposed the inclusion of categories such as social status, familial and social position, life histories, and the specificities of sociocultural contexts, including rurality, to attain a better understanding of WF dynamics. Other studies have supported this proposition. For instance, in Mexico, Ayala-Carrillo and Peréz-Fra [20] emphasized the experiences of Indigenous artisan women who engage in both artisanal and domestic work, with the former adopting characteristics similar to the latter in terms of lower value and inadequate remuneration. In Peloponnese, Greece, Anthopoulou [21] demonstrated how the determination of farming women to achieve success is influenced by the need to balance their work in the food industry with their family responsibilities. Likewise, in the rural context of the United States, Son and Bauer [22] revealed the necessity for single, working, and low-income women to establish a work–life balance in order to achieve autonomy and professional growth. Therefore, these studies have highlighted the significance of women’s advancing in terms of WF balance to gain social and occupational recognition whilst furthering their decision-making opportunities and improving their ability to meet their own and their families’ needs.

### 1.2. Work–Family Interface in a Farming Setting

Contzen and Forney [23] and Dunne, Siettou, and Wilson [24] analyzed the rural family structure in Western and European countries. They highlighted that, although men were traditionally the farmers and farm managers and women the helpers who supported their husbands by taking care of young cattle, hens, gardens, households, and families, new forms of labor division are emerging. Farming women often develop individual professional careers in and outside of farming and organize themselves in order to gain more free time all whilst facing growing rates of divorce [23,24]. At the same time, a mix of agricultural policy to support farmers and a progressive deregulation of the market, along with climate variability, the times for the production of goods, the seasonality of production, the geographical separation between production regions and end users, frequent natural disasters, price uncertainty, imperfect markets, and a lack of financial services, such as credit and insurance, are particular stressors for farmers, making daily life increasingly difficult [23,25].

In the USA, Sprung and Jex [5] found that male farmers with higher job engagement reported better psychological health and fewer physical symptoms while, for their wives, farm satisfaction was positively related to psychological health and negatively related to physical symptoms; however, women farmers’ job engagement was not monitored in the study. In Canada and Australia, Wendt and Hornosty [26] showed that health services in rural areas relegate gender violence and women’s illness to the background, with partners and children but not women being the first to receive health care. In East Africa, Hyder et al. [27] analyzed women’s work in food collection, housework, and care tasks and its social and health implications on women. Neither of these studies delved into the WFI of women farmers.

Rural areas are characterized by unique socioeconomic and environmental factors that significantly impact the health of women, representing significant challenges for many states. Rural women also face considerable challenges in accessing adequate healthcare services, including primary care, specialty care, and reproductive health amenities [28,29]. Limited access to healthcare exacerbates health disparities and increases the likelihood of undiagnosed conditions. It has been reported that rural women experience higher rates of chronic diseases compared to their urban counterparts. Research by Hartley et al. [30] demonstrated that rural women have an increased prevalence of obesity, hypertension, diabetes, and cardiovascular diseases. Additionally, the rural environment presents unique mental health challenges for women. Studies have shown that rural women experience higher levels of depression, anxiety, and stress due to social isolation, limited social support networks, and increased financial strain [31]. Additionally, the stigma associated with seeking mental health services in small, tight-knit communities further impedes timely access to appropriate care [32]. No less important is the fact that rural women often engage in physically demanding occupations, such as agricultural work, which exposes them to various occupational hazards. These hazards include exposure to pesticides, chemicals, and physical injuries, which can have detrimental effects on their health [12]. Occupational health and safety regulations in rural areas are often less stringently enforced, further contributing to increased risks and limited protection for women.

Research has shown not only reduced access to resources in terms of medical assistance and infrastructure but also a medical community limited in their knowledge about rural lifestyles and the territorial and sociocultural determinants of health and disease [33]. Also, Duclos et al. [34] and Chen et al. [35] showed the relevance of population knowledge in formulating public health policies in order to strengthen relationships between users and professionals regarding attention, needs, and information. Addressing these issues requires a comprehensive approach involving improved healthcare infrastructure, targeted interventions for chronic disease prevention and management, increased mental health support services, and enhanced occupational safety measures. Thus, the inclusion of the WFI from a farming women’s perspective could improve equity, accessibility, and effectiveness in the development of public policies and the exercise of health services [36].

In this context, our research attends to the rural WFI gap from the point of view of farming women’s experiences. Our study aimed to analyze how the characteristics of farming women’s work and family organization are related to the WFI. Specifically, we asked which facilitators and stressors are involved in WF dynamics and their consequences for women’s physical, psychological, and social well-being. We focused our study on Chile, where 13.7% of the population was rural in 2002 and 11.4% in 2021 [37]. Additionally, the percentage of rural households headed by women increased from 14.0% to 26.5% between 1990 and 2013; such households require more health services, job opportunities, and transport services [38].

## 2. Methodology

### 2.1. Design

We used a qualitative method with an analytic scope and a gender perspective [39] to analyze WF relations in the context of farming families from women’s point of view. We addressed this objective using a grounded theory design with a constructivist approach [40,41] that sought to build a theoretical framework for the object of the study.

### 2.2. Setting

We carried out the study in the region of Araucanía, Chile, where the economic participation of women is a key human resource for agriculture, livestock rearing, and fishing [38]. The contribution of farming families to the overall economic development of the country is substantial. Specifically, they contribute to national vegetable (54%), crop and flower (40%), wine grape (30%), goat (94%), honey (76%), and cattle (54%) production [42].

According to the 2017 census, the Araucanía region was classified as having a high rural territorial extension since the municipalities of this region are predominantly rural (nineteen), while only two are classified as predominantly urban and eleven as mixed. Together, the mixed and rural municipalities cover 94.8% of the regional territory [37]. Only one in three homes has access to the public water network, less than 5% have no electricity, and only one in 10 has internet access [37].

### 2.3. Selection of Participants

Our sample was non-probabilistic, based on the maximum variation technique aimed at varying the characteristics of informants working in a diversity of situations [43]. The criteria of inclusion were: (a) being a farming woman, (b) playing the role of a housewife, and (c) being a resident of Araucanía. As variability criteria, we considered the following: (a) participating more or less in economic or community activities and (b) the type and quantity of care demands (e.g., children under five years old or elderly, non-autonomous people).

Regarding the first variability criteria, we used the categories proposed by Dune et al. [24] to classify farm women according to the economic contribution and visibility of their work. We, thus, focused on the following three categories: traditional housewives, familial workers, and farmer–workers. A traditional housewife (equivalent to a *traditional farm housewife*; Dune et al. [24]) assumes the responsibility for housework and caring for the family and eventually helps her husband around the farm. When she earns an income, she subjectively recognizes it as her husband’s, rather than her own, so her economic contribution is rendered invisible to herself. On the other hand, the familial worker category refers to farm women who describe their own economic contributions as being of assistance to a male farmer. Like traditional housewives, a familial worker is responsible for domestic and care chores, but she is also responsible for suitable farm tasks (e.g., labor for youngstock/livestock) or even for minor farm enterprises (e.g., sales). A familial worker is, thus, similar to a *farm assistant*, according to Dune et al. [24].

Finally, the farmer–worker represents all women who assume the responsibility for domestic, care, productive, and eventually communal activities. Usually, their productive or communal tasks are carried out at home and require them to become involved in training and acquiring public funding. Such workers pursue the growth of the farm, generate personal money, and can eventually become temporary employees. Some may have studied a profession or a technical career. Therefore, farmer–workers are similar to the transition in the categories of Dune et al. [24] from *subordinate managers* to *traditional women farmers*.

### 2.4. Data Collection

Semi-structured interviews were designed as a data collection technique and complemented by field notes. All the interviews were transcribed verbatim. Interviewing took place between 2019 and 2023. The majority of the interviews were carried out on location, but during the COVID-19 pandemic, some were conducted by phone or implemented on location by following the social restrictions to prevent COVID-19 infections. Throughout the research period, the informants were encouraged to participate in scientific dissemination activities or to take part in other educational or research projects for rural development.

### 2.5. Analysis

Plan analysis was procedurally structured from grounded theory, following the steps of open coding, line by line, axial coding, and selective coding [40]. The information from the interviews was systematized based on the emerging categories, and then the information was coded to establish levels of depth in the analysis [44]. This process involved the constant review and comparison of the data that were being obtained in order to build an emerging theory [44], in this case, on the link between WF reconciliation and the configuration of farming work in rural areas. The categories of the emerging theory were ordered as shown in results.

### 2.6. Methodological Rigor

To ensure methodological quality and rigor, the coding process was performed by two members of the research team whose professional profiles were complementary: one of them is an agronomist, and the other is a community psychologist with extensive experience in gender studies. In this way, it was implemented as a triangulation process of researchers who interchanged theoretical concepts and perspectives from which to enrich the results of the research [45]. Also, the preliminary results were triangulated with information gained from 12 technical experts [46], mostly agricultural, social, and health professionals and technicians, who worked in rural areas of Araucanía and who provided information about the social and gender contexts of interviewees. Finally, the bibliographic information was reviewed by experts to ensure its reliability.

### 2.7. Ethical Considerations

This study was approved by the Ethics Committee of the Catholic University of Temuco. The applicants were given a consent form that informed them that their personal information would be safeguarded. The real name of each participant was replaced with a number in order to preserve confidentiality, such that no information could be used to identify informants.

## 3. Results

Table 1 shows the composition of the final sample. The forty-six participants were four traditional housewives, with these women being between 33 and 49 years of age and their age range being less variable, sixteen housewives who were family workers, with these women being between 24 and 86 years of age, and twenty-six housewives who were agricultural workers, including women between 22 and 60 years of age. The type and quantity of care demands concentrated on caring for children and young people (under 18 years old). We managed to vary the sample by adding two interviewees who were responsible for caring for elderly or dependent adult people (e.g., a husband or parents). At least 11 informants had sons and daughters who represented a lower care load (e.g., lived away from home).

### 3.1. Family Farming Load Distribution

According to the informants’ perspectives and as shown in Table 2, their familiar load distribution could be understood through the criteria of gender role distribution. This gender role distribution took form according to the typical familial roles of the head of the family, the housewife, and the children for every member of the family. Adult men are the heads of the families, are responsible for productive activities, and are mostly remunerated. Children assume assistance responsibilities; boys help by cutting and carrying wood, while girls help by cooking, cleaning, and often taking care of younger children. They integrate these activities as part of their daily lives, as most children receive an education. The characterization of women’s work is based on their roles as traditional housewives, familial workers, or farmer–workers, according to the degree of productive and community tasks they perform. As most informants were familial workers and farmer–workers, this means that they met the growing demands of work. Additionally, we described whether the productive, reproductive, and communal activities were undertaken outside or inside the house.

### 3.2. Facilitators of and Stressors for Women’s Balance and Conflict in Family Farming

We identified the facilitators and stressors according to whether they emerged as a balance or a conflict, respectively, between work and family in the experiences of the women. In Table 3, we show the rural feminine balance and conflict, characterized as a process depending on whether women express conformity or nonconformity with their workloads. When women feel they are able to balance work and family demands, even if they have a multitude of daily tasks, they express conformity with their roles and understand that they are “multitaskers”. Because of this, they are able to undertake community, productive, and reproductive work simultaneously or in a spatially integrated manner. In this strategy of concealment, these women frequently work outside, whilst their children assist them, or they work indoors, for instance, processing small-scale forestry products, whilst caring for others. Other facilitators of the balance are having access to basic services, being well-trained for their productive jobs, having social support, and having help from nearby families in caring for children and young people.

Eventually, the women felt that they were able to fulfill their work responsibilities but with high emotional, labor, and physical costs. At that point, individuals expressed an emerging dissatisfaction and experienced WF conflicts. This typically occurred in the presence of the stressors identified. Such strains included dedicating extra time to carrying potable water home, poor training for their productive jobs, or facing special situations, such as having to care for children under 5 years old or for individuals with illnesses or elderly family members. Other stressors happened when youths emigrated to the city and women had to assume their work responsibilities or when a new neighbor arrived and women did not trust him/her to work together in common activities (for example, to improve roads). However, when the women experienced such conflicts, they did not feel that they had the right to express disconformity or distract themselves with a leisure activity.

### 3.3. The Consequences of Work–Family Balance and Conflict on Women

Table 4 shows that the balance/conflict process has physical, psychological, and socioeconomic implications for women. Women experience these consequences in different ways, depending on whether they are in the balanced or conflictive stage. The main difference is that negative effects on health are less or more visible for women, respectively. But, even if they are experiencing a conflict characterized by an emerging disconformity, women show little awareness of their biopsychosocial distress.

When women consider themselves balanced, even though they may report exhausting journeys and constant time pressures, they feel proud and satisfied, as they are busy all day, are earning money, and are constructing new opportunities for future generations. In this way, their high workloads are invisible to them. On the other hand, when women declare that they are strained, they identify the negative consequences of their exhausting journey and may feel drained, sore, angry, helpless, trapped in monotony, and in poor humor. Also, they have insufficient time for all their essential activities and experience difficulties in organizing visits to health and social services and training. In the most extreme cases, a husband may even prohibit his wife from becoming a worker or a social-community leader.

## 4. Discussion

### 4.1. Family Farming Load Distribution

The load distribution in family farming is the structure through which facilitators and stressors play a role in the emergence of a balance or a conflict between work and family in the experience of women. Because family farms function by distributing the load (tasks) between all members, including children, women are able to fulfill all their care and productive tasks, including those in their community, often with the assistance of their children. On the other hand, except for the head of the house, the farming load entails multiple roles for each family member, such that women and sometimes children may have to adapt to balance productive and reproductive tasks in a way that blurs the boundaries between work and family both spatially and temporally.

This kind of labor organization produces many effects regarding women’s time and work. Firstly, their time expands and becomes enough to fulfill all their activities even if they are involved in multiple duties, given that women often also perform essential tasks in their rural communities. However, women’s time becomes devoted to their families but not to themselves to such an extent that many women do not have time to feel distressed, relax, enjoy leisure activities, or attend to health and other personal services. Such a lack of the ability to access these services was a feature expressed by all the farming women (farm workers, familial workers, and housewives), and it is consistent with previous reports. Indeed, Rodríguez and Muñoz [47] and Shui et al. [48] expressed that the organization of rural farm work creates an illusion of excess of time for women, forcing them to be multitaskers and to work harder in order to gain self- and social validation.

### 4.2. Facilitators of and Stressors for Women’s Balance and Conflict in Family Farming

In Table 3, we identify that being a multitasker versus emerging disconformity are the key outcomes of determining the WF balance or conflict, respectively, of farming women, and we explain the main facilitators and stressors that contribute to this process. Consequently, we suggest that, when farming women feel in balance, there is actually still a latent tension because they have multiple duties that are time- and energy-intensive, which could create personal and familial conditions that subsequently emerge as a WF conflict. Also, we highlight that gender-based cultural beliefs and roles facilitate the presence of this balance. In fact, women often feel proud of their strong capacity to work on multiple fronts at once (even more so than their husbands), they are satisfied by their capacity “to be a multitasker,” and they try to spatially and temporally integrate their domestic, care, and productive tasks. We found this kind of performance among women in both rural and non-rural contexts, but the boundaries between the domains are often blurred. For example, Ref. [49] showed that female entrepreneurs in Ethiopia use integration as a work–life boundary management strategy, understanding integration as opposed to segmentation or alternation between domains.

We have identified specific family and community situations and classified them as facilitators or stressors of WF balance/conflict, distinguishing them according to their context (institutional, communal, or family) or whether they are interrelated with gender roles. We imagined that striking a balance for farm women would mean invisibilizing any WF conflicts that could erupt when the precarious family organization weakens, allowing stress factors to dominate. Nevertheless, the main obstacle that prevents the emergence of an obvious WF conflict and that makes it almost impossible to identify the direction of the potential conflict (work-to-family or family-to-work) is due to the gender and temporal–spatial characteristics of the farming workload of women.

Our results partially agree with other WFI studies that have focused on the experiences of employees. Factors for employees, such as long working hours, extended commuting times, being a multitasker, and having children under the age of five, are also stressors for farming women [4]. In the same way, family support and being married are facilitators of a WF balance among farming women [4]. Nonetheless, our results indicate significant distinctions in the mechanisms through which these factors function. Specifically, farming family support encompasses not only emotional and financial aspects, as examined by Welsh and Kaciak [50] in the context of businesswomen, but also the involvement of physical labor. Furthermore, this support extends beyond adults to encompass children and young individuals.

### 4.3. Consequences of Women’s Work–Family Balance/Conflict

In Table 4, we evaluate how the balance/conflict between work and family domains has physical, psychological, and socioeconomic implications for women. The WF relationships in the experiences of farming women have very positive psychological consequences that paradoxically facilitate a balance that is precarious. On the one hand, some facilitators of agriculture, such as spatial and temporal integration, and some tasks, such as farm care, could decrease stress symptoms [7,51,52]. These kinds of activities have been tested by health professionals in order to diminish depression rates and improve the health of the elderly population [53,54]. Also, being a solo-preneur or working on a farm facilitates women in spending less time traveling to workplaces. So, indirectly, it reduces mobility barriers. Other studies have shown that farming women experience wellbeing and personal satisfaction when they are able to fulfill their daily tasks because it is a way to be socially recognized as a “good housewife” and have even related these feelings to subjective empowerment and an increase in female autonomy [55]. In other words, women’s decision-making processes appear to be marked not only by their purpose of being autonomous people but also by their being good farming housewives. However, farming work generates an all-day work demand and usually results in chronic distress, time poverty, psychological difficulties in disconnecting from work, and obstacles to accessing a better quality of life.

The WF balance on farms could result in unforeseen consequences because women tend to suppress their physical and socio-psychological distress. The organization of a farming load encourages women to muffle their right to express pain and bad feelings, which also reinforces traditional gender-based cultural patterns by highlighting the fact that women feel that they are without the right to express disconformity with their situations in the context of WFI theory. This constant is the lack of access or consultation in health services among women. The feeling of dissatisfaction with peasant family farming translates into loneliness and personal problems for women. Our results improve the knowledge about WF relationships and their consequences for women. For instance, Wendt and Hornosty [26] in Canada and Australia and Hyder et al. [27] in East Africa have shown that women’s health issues are relegated to second place in comparison to the medical situations that other familiar members are affected by. Our findings confirm that it is necessary to understand the organization of family farms and their potentially harmful impacts on women, who traditionally have occupied the social position with less power [8,23].

According to our findings, there is a prejudice that women “enjoy” an excess of time, which exposes them to multitasking. Left unchecked, this prejudice means that many farming women never have time to access health services. Additionally, we revealed that the increase in stressors is associated with an increased risk of physical illness or psychological exhaustion. These findings are in line with those described by Artazcoz et al. [56], who found that time limitations mean that women are more prone to illnesses going undiagnosed and score poorly in health indicators. On the other hand, and as reported elsewhere, we found that the strengthening of community networks allows such negative effects on women to become cushioned [57].

In summary, farming women’s labor and family positions could favor two relevant WFI dynamics: firstly, that women have little opportunity to express to others that the WF dynamics they are involved in have consequences on their health and social opportunities, and secondly, that women only are able to recognize a WF conflict when they become sick, feel extreme loneliness, or face other serious family difficulties such that they become unable to fulfill productive and household roles.

## 5. Conclusions

Our study aimed to explore the stressors and facilitators involved in WF dynamics and their impact on the physical, psychological, and social well-being of women engaged in farming activities. To achieve this, we employed a qualitative method with a grounded theory design, conducting semi-structured interviews with 46 women farmers from the Araucanía region in Chile. According to our findings, the family farming load distribution shapes the work and family experiences of women. The workload is shared between family members, allowing women to fulfill care, productivity, and community tasks. Multitasking roles are common for women and children in family farming, blurring the boundaries between the work and family domains. Women’s time expands to accommodate their multiple responsibilities, including communal tasks. However, their time becomes predominantly focused on family and community needs, leaving little or no personal time for leisure or accessing personal services. This lack of personal time is an experience shared by many farming women. Cultural beliefs and gender roles contribute to women’s sense of balance, as they take pride in their high work capacity and strive to integrate their domestic, care, and productive tasks. Specific family and community situations can facilitate or stress the WF balance, depending on their context and relation to gender roles. Achieving balance often conceals an invisible WF conflict that emerges when the familial organization is disrupted by stressors. These situations are guided mainly by women’s feelings of dissatisfaction with the new organizational dynamics in peasant family farming. Stressors are generators of anguish and loneliness in women, who understand that they affect their well-being and discrimination in consulting with health services. The gender and temporal–spatial characteristics of women’s workload make it challenging to identify the eventual direction of a conflict.

The WF balance or conflict in family farming does indeed have physical, psychological, and socioeconomic implications for women. While women feel rewarded and socially recognized for their all-day engagement, this balance often leads to chronic distress, time poverty, and barriers to a better quality of life. In this work, we propose the existence of a prejudice that implies that women have an excess of time, which often leads to increased multitasking. This constant demand for multitasking directly interferes with women’s access to healthcare and their overall well-being. Furthermore, we suggest that the accumulation of stressors associated with multitasking is linked to a higher risk of physical illnesses and psychological exhaustion. However, our research also indicates that the strengthening of community networks can help mitigate these negative effects on women’s health.

This research shares a limitation with the vast majority of studies on the health and the social and economic participation of rural women—that is, there is no data about the masculine partners of participants. By collecting data from the perspectives of the latter, scholars could advance by identifying changes in the perceptions of men about their female partners and the opportunities they open for gender equity. We encourage scholars to research these ways. As far as we know, this is the first study which investigates both WF facilitators and the WFI, considering gender differences and the influence of sociocultural environments beyond formal employment organizations. More research is needed to examine the stressors and outcomes related to the interplay between work and family responsibilities, primarily focusing on the perspective of employed individuals. In this sense, it is relevant that future research can deepen the relevance of the personal and environmental factors that generate stressors and facilitators in work–family balance. This must be highlighted with the diversity of contexts and roles from which women operate to enrich and give complexity to the phenomena of the WF balance. Investing in health programs and primary care with this focus should mitigate the negative effects of multitasking on women’s health. By offering time management support, fostering community engagement, and providing accessible and comprehensive care, women’s well-being can be improved. Emphasizing social support networks and addressing both physical and mental health needs are crucial. These measures can help women better manage time, reduce health risks, and promote overall health and well-being.

## Figures and Tables

**Table 1 healthcare-11-02726-t001:** Final sample composition.

Informant Work Roles	Number of Participants	Type of Care Demand	Number of Participants
Traditional housewife	4	Children under 18	4
Familial worker	16	Children under 18	8
		Other conditions (e.g., adult sons and daughters who lived away from home)	6
		No data	2
Farmer–worker	26	Children under 18	17
		Elderly people	2
		Other conditions (e.g., adult sons and daughters who lived away from home)	2
		No data	5
Total informants	46		46

**Table 2 healthcare-11-02726-t002:** Gender work distribution of farming families.

Criteria	Familiar Position				
	Head of Family	Housewife			Assistant
		Farmer–Worker	Familial Worker	Traditional	
Gender work distribution	Productive (salaried/agricultural) and community (leisure)	Reproductive (care and domestic), productive (entrepreneurship and agricultural), and community (community organizations and institutional participation; not leisure activities)	Reproductive (care and domestic), productive (requirements of the head of the family), and community (care for children)	Reproductive (“I don’t work”)	Boys: reproductive (taking care of younger children, mainly when there is not a sister) and productive (collecting firewood, watering plants, and other lighter agricultural tasks) Girls: reproductive (taking care of younger children) and productive (food preparation, shelling peas, and other lighter agricultural tasks)
Outside/inside work	Mostly outside the house	Outside and inside the house	Mostly inside the house	Mostly inside the house	Mostly inside the house
Testimonials	“In the case of my father, he hardly did anything in the house” (I8) “They’re dedicated to their crops, they get together, work the land, chop wood; that’s what I’ve seen the menfolk doing” (I7)	“Eucalyptus plantations” (I43) “Milking” (I51) “You still have to support the household budget, because if farming is your livelihood, that’s where most money can come from” (I4)	“At home, make lunch, bake bread, wash, attend the animals” (I1) “[I work] on the vegetable patch…collecting seeds, making jams” (I4)	“At home, I do everything, housework, making lunch” (I6) “Mothers are in charge of caring, whilst fathers provide” (I3)	“The eldest looked after the youngest and helped mum” (I35) “I try and teach the basics, like counting chicks, additions, so that the kids don’t lose their intelligence” (I34) “For example, my mother taught me to bake bread when I was 6 years old” (I4)

**Table 3 healthcare-11-02726-t003:** Women’s balance and conflict in family farming.

Sphere	Work–Family Relations			
	Balance: Being a Multitasker		Conflict: Emerging Disconformity	
	Facilitators	Facilitator Testimonials	Stressors	Stressor Testimonials
Institutional context	Access to basic services (drinking water, electricity, roads, and public buses) Training (to be a more competent worker)	“[With technology], now the farm women look after the greenhouse, such that they work all year round” (I10) “I used to be in a technical advisory that was always applying for subsidies” (I45)	Limited or no access to basic services (no drinking water or electricity, poor roads, or limited public buses) Training (schedules are not compatible with family care necessities)	“There is no possibility of irrigation” (I45) “The internet [during the pandemic]… there wasn’t much information [generates uncertainty]” (I2)
Community	Community care work (grandmothers)	“My son never went to nursery as I had the help of my parents and I prefer to have him with them rather than take him there” (I4) “Even the grandads [take care of the grandchildren], the grandmas help so their daughters can work” (I6) “Grandmothers look after their grandchildren, because these days, grandmas are younger” (I35)	Social mistrust (e.g., recent urban-to-rural migration)	“I don’t talk with my neighbor here” (I46) “Here people are so bad, you can’t trust anyone” (I6) “Now an indigenous community has arrived here, [there are] several communities and they are conflictive” (I41)
Family farming workload organization	Gender and age work distribution Having paid workers	“Because a woman has to do more things [compared to a man]” (I35) “Daughters are taught [to help], whilst the fathers take the sons out to work with oxen, and things like that” (I1)	Children under 5 years old Youth migration Becoming a widow or divorcée Caring for sick and/or elderly adults Having cash to hire workers	“The children leave and you are left alone, you have to fight with what you have” (I1) “Since the kids don’t help these days, one has to hire people” (I35) “Look, my dad spent 8 years with a disease, cancer, 6–7 months bedridden, at 7 months he passed away” (EI)
Gender	Spatial and temporal integration	“Women know how to cope with a lot of work” (I3) “Women plan the day so that they have enough time to do their jobs. […] And it strikes me that, on the other hand, a man is always doing what he has to do, then he rests; instead, while the woman works, cooks, she is doing other things” (I34)	Without the right to express pain or disconformity Without the right to leisure	“Out of nowhere, as a housewife, you get tired, but you have to continue as if nothing has happened” (I3) “A farm woman has no fun” (I10)

**Table 4 healthcare-11-02726-t004:** The consequences of work–family balance and conflict on women.

Sphere	Work–Family Relations			
	Balance: Being a Multitasker		Conflict: Emerging Disconformity	
	Consequences			
	Invisible	Testimonials	Visible	Testimonials
Physical	Exhausting journey	“24 h a day busying around [working]” (I10) “We women know how to cope with suddenly being overworked” (I3)	Becoming exhausted Diffuse pain	“Countryfolk have more work and are more worn out physically, the pain in the bones, and all those [diseases] that comes now [with old age]” (I36) “It’s hard work, in the afternoon, one is exhausted” (I7)
Psychological	Proud of being busy all day (“Being a good housewife,” “being competent for doing everything”) Proud of having a job or earning money (only family workers and farmer–workers) Satisfaction in constructing new opportunities for future generations and communities	“[Work] helps you feel useful, agile, to feel good, so you don’t feel down, you feel good because you know you have to get up […] you feel good, you’re filled with more energy” (I5) “My daughter is still studying, she has two years left to finish her [university] degree, so I’m happy for her” (I9)	Angry Feeling of helplessness Disempowerment Feeling of monotony Poor humor Lack of time (time poverty) Gender-based violence Feeling guilty	“Anger, helplessness, that one can do the chores, I’m in a bad mood” (I2) “I think I failed [as a mother] when I was young and many [affective] gaps remain, that are still present when the children are older” (I68)
Socioproductive	Squeezing the most out of her time	“We are multifaceted, we have to get so much done, time is very short” (I2) “Just imagine the women, I get up at 6:50 in the morning, I go to bed almost at midnight, so time for myself, for earning, is short” (I4)	Difficulties in organizing access to benefits and general services (for example, health and social) Difficulties in getting training in productive and technological areas	“I cannot take part in meetings, train very occasionally, because there is no one at home to do things for me” (I45) “When do we have time to do such paperwork [if we are busy all day]?” (I42)

## Data Availability

Not applicable.

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
