# Peer review of "Uncovering the Work–Family Interface: The Impact of Facilitators and Stressors on the Health of Farm Women"

_healthcare, 2023, doi:10.3390/healthcare11202726_

Round 1

Reviewer 1 Report

Thank you for the very well written and interesting article on the work-family interface in relation to health among farm women.

Overall, I have only a few comments, which will please be taken into consideration in the revision.

Methodological notes:

I would ask you to review the text in general to ensure that all relevant methodological aspects have been taken into account:

For help, please use one of the established guidelines for reporting quantitative studies.

This can be found on the Equator Network website – https://www.equator-network.org/reporting-guidelines

The SRQR guideline with checklist is useful. For example, the following remarks should still be added to the text.

From line 242: Data Collection: Here it is not clear how the recruitment was done or how the women were asked to participate in scientific activities. Were the potential participants contacted or recruited by phone? Please provide additional information.

How many interviewers were conducting the interviews. Were the interviews recorded (dictaphone)? Transcription - software? Duration of the interviews? Min-max / average.

From line 249: Analysis: Was any software used in the analysis to support the process? If so, please elaborate on this here.

From line 337: Discussion: Limitations or strength/weaknesses of the study are not explicitly discussed. Please add.

Minor error revisions in the text.

Line 189: Correct to "in Chile".

Lines 210 to 212: Please correct the sentence.

Author Response

Dear Reviewer, 

We are very grateful for your comments which helped us perfect our work. All were resolved in the attached document

Author Response

(The authors gave the same response as above.)
